# Dermoscopy of Actinic Keratosis: Is There a True Differentiation between Non-Pigmented and Pigmented Lesions?

**DOI:** 10.3390/jcm12031063

**Published:** 2023-01-30

**Authors:** Dimitrios Sgouros, Melpomeni Theofili, Theodora Zafeiropoulou, Aimilios Lallas, Zoe Apalla, Alexios Zaras, Konstantinos Liopyris, Georgia Pappa, Eleni Polychronaki, Fiori Kousta, Antonios Panagiotopoulos, Alexander Stratigos, Dimitrios Rigopoulos, Alexander C. Katoulis

**Affiliations:** 12nd Department of Dermatology and Venereology, “Attikon” General University Hospital, Medical School, National and Kapodistrian University of Athens, 12462 Athens, Greece; 21st Department of Dermatology and Venereology, “Andreas Sygros” Hospital, Medical School, National and Kapodistrian University of Athens, 16121 Athens, Greece; 31st Dermatology Department, School of Health Sciences, Aristotle University of Thessaloniki, 54643 Thessaloniki, Greece; 42nd Dermatology Department, School of Health Sciences, Papageorgiou General Hospital, Aristotle University of Thessaloniki, 56429 Thessaloniki, Greece

**Keywords:** skin tumors, dermoscopy, non-pigmented actinic keratosis, pigmented actinic keratosis

## Abstract

Dermoscopic features of actinic keratosis (AK) have been widely studied, but there is still little evidence for their diagnostic accuracy. Our study investigates whether established dermoscopic criteria are reliable predictors in differentiating non-pigmented actinic keratosis (NPAK) from pigmented actinic keratosis (PAK). For this purpose, dermoscopic images of 83 clinically diagnosed AK (45 NPAK, 38PAK) were examined, and the sensitivity (Se), specificity (Sp), positive predictive value (PPV), and negative predictive value (NPV) were assessed. Features with statistical significance were the red pseudo-network (*p* = 0.02) for NPAK and the pigmented pseudo-network (*p* < 0.001) with a pigment intensity value even less than 10% for PAK (*p* = 0.001). Pigmented pseudo-network (Se: 89%, Sp: 77%, PPV: 77%, NPV: 89%) with a pigment intensity value of more than 10% (Se: 90%, Sp: 86%, PPV: 79%, NPV: 93%) had excellent diagnostic accuracy for PAK. Scale and widened follicular openings with yellowish dots surrounded by white circles were equally represented in both variants of AK. Linear wavy vessels and shiny streaks were more prominently observed in NPAK, as were rosettes in PAK, but these results failed to meet statistical significance. The red starburst pattern was near statistical significance for PAK. Therefore, pigmentation is the strongest dermoscopic predictor for the differentiation between NPAK and PAK.

## 1. Introduction

Actinic keratosis (AK) is a premalignant lesion within the cutaneous squamous cell carcinoma (cSCC) spectrum, with a prevalence ranging from 11% to 60% in Caucasians above the age of 40 years and a potential for malignant transformation into invasive cSCCs ranging between 0.1–20% [1,2,3,4,5,6]. It commonly occurs in elderly, fair-skinned male individuals, in areas of sun-damaged skin [1,2,5]. Non-pigmented actinic keratosis (NPAK) and pigmented actinic keratosis (PAK) consist of the main clinical subtypes of AK defined by the absence or presence of clinically obvious pigmentation, respectively [2].

Dermoscopy is a non-invasive examination that improves the clinical diagnosis and early and efficient management decisions for skin lesions [7]. In addition, a dermoscopic examination contributes to its treatment through the preprocedural assessment of lesion surface boundaries, the monitoring of the effects of local therapies, and the long-term monitoring of patients [6,8].

Dermoscopically NPAK facial lesions are characterized by the so-called “strawberry pattern” consisting of a red pseudo-network created by the background erythema with targetoid ostia of hair follicles (white circles corresponding to the follicular openings, surrounding a yellowish keratotic plug) [2,9,10]. On the other hand, PAK lesions of the face demonstrate a pigmented pseudo-network consisting of a pigmented background between targetoid follicular openings, at times associated with a red pseudo-network [11,12]. Additional dermoscopic features include a white-to-yellow surface scale, fine wavy vessels surrounding the hair follicle, and structures only visible in polarized-light dermoscopy. These include a rosette sign (four white, dotted clods forming a four-leaf clover-shaped structure, due to alternating hyperkeratotic and parakeratotic corneal layers in the follicular infundibula with peri-follicular fibrosis) and shiny streaks (white, perpendicular, few-millimeters-long lines caused by the polarization of thickened hyaline fibrous bundles) [9,12,13,14]. The progression to malignancy of the AK is highlighted by the “red starburst pattern”, radially arranged red lines, or hairpin vessels surrounding a yellow-to-white scaly center [2,15].

Our study aims to improve the clinical differentiation between NPAK and PAK of the face and scalp through an evaluation of the diagnostic accuracy of the dermoscopic criteria of AK, as PAK shares clinical and dermoscopic features with Lentigo Malignant (LM). The differential diagnosis of PAK from LM sometimes represents a challenge even for expert dermatologists, and a histopathological examination is important before starting treatment of this variant of AK. Thus, it is highly significant to describe all the aspects of the dermoscopic morphology of PAK.

## 2. Methods

### 2.1. Study Population

We included 40 patients with 83 clinically diagnosed naive AK from the Dermato-Oncology Unit of the 1st Department of Dermatology & Venereology of “A. SYGROS” University Hospital in Athens, Greece. The study was approved by the Ethics Committee of the institution (approval code 2530/5,10.2018). All participants were informed about the aims of the study and provided written informed consent. All lesions included in our study had typical clinical and dermoscopic features of AK, and the evaluation of their criteria met an agreement between two independent investigators (DS and AK). In case of differential diagnostic doubt, a skin biopsy was performed and lesions that did not have a definite AK histopathologic diagnosis were excluded from the study. AKs arising on extra-facial anatomic sites were not included in the study. Another exclusion criterion was actinic lesions that were previously treated for AK irrelevant of the therapeutic method.

### 2.2. Clinical and Dermoscopic Examinations

Initially, the demographic characteristics of the patients were recorded, followed by a clinical examination. AK lesions were clinically divided into two categories for the purposes of the study: either NPAK defined as lesions without clinically obvious pigmentation, or PAK that showed even light, clinically observed brown–black coloration. Moreover, AK lesions were clinically classified according to the Olsen’s system. Lesions better identified by palpation than by inspection were defined as Olsen grade I, lesions easily identified by inspection and palpation were Olsen grade II, while hyperkeratotic lesions were classified as Olsen grade III [16]. Dermoscopic images were obtained using polarized-light, contact dermoscopy, with Dermlite DL 200 Hybrid dermoscope coupled to a Nikon J1 camera. Their evaluation was performed for the predetermined dermoscopic criteria of AK (red pseudo-network, pigmented pseudo-network, scales, yellowish dots within widened follicular openings, white circles around yellowish dots, linear wavy vessels, short shiny streaks, and red starburst pattern). 

In addition, the intensity of the pigmentation was measured as the percentage of the surface of the lesion covered by the pigment. According to the intensity of the pigmentation, lesions were classified into 4 categories: lesions without pigmentation, lesions with pigment intensity less than 10%, lesions with pigment intensity of 10–50%, and lesions with pigment intensity greater than 50%. 

### 2.3. Statistical Analysis

The statistical analysis was performed with STAT/IC version 15.1. Continuous variables were checked for normality of distribution with the Shapiro–Wilk test and were presented as mean ± standard deviation. Contingency tables were used for categorical variables. For the detection of statistically significant differences in the presence of dermoscopic criteria between NPAK and PAK, as well as for the rest of the categorical variables, Pearson’s chi-squared and Fisher’s exact tests were used. For continuous variables, Mann–Whitney and Kruskal–Wallis tests were used. Additionally, for the statistically significantly different variables, univariate and multivariate logistic regression analyses were conducted. Finally, in the multivariate logistic regression model, an evaluation of the goodness of fit was performed according to the Hosmer–Lemeshow criterium. Statistically significant differences were considered in cases of *p*-value < 0.05.

## 3. Results

### 3.1. Demographic and Clinical Features of the Study’s Population

In total, 45 NPAK and 38 PAK were diagnosed in 40 patients. Of them, 18 patients presented with a solitary AK lesion, 10 patients with 2 lesions, 6 patients with 3 lesions, 4 patients with 4 lesions, 1 patient with 5 lesions, and 1 with 6 AKs. No major differences were noted regarding the age and sex of our study participants in both types of lesions with AK. Patients with NPAK and PAK tended to be older, with a predilection for the male sex [NPAK: 38/45 (84.4%), PAK: 31/38 (81.6%)] (Table 1).

Concerning the Fitzpatrick skin-type classification, the most frequent phototypes were II and III for NPAK and PAK (II: ⅓ in both, III: ¼ of NPAK, IV: ½ of PAK patients) (Table 1).

Regarding the anatomic site of the lesions, common sites of presentation were in NPAK lesions, the scalp in 18/45 (40%), and the lateral face in 9/45 (20%). PAK lesions were approximately equally present on the scalp, temples, and lateral face (26.3%, 26.3%, and 28.9%, respectively) (Table 1). 

Furthermore, the macroscopic examination of the lesions revealed that the majority of NPAK and PAK lesions were OLSEN’s stage II (18/45 (40%) and 18/38 (47.4%), respectively, followed by stage I (16/45 (35.6%) and 13/38 (34.2%), respectively (Table 1).

### 3.2. Dermoscopic Features of NPAK and PAK Lesions

The dermoscopic features of NPAK and PAK are presented in Table 2. With regard to color, the red network was more frequently found in NPAK lesions [38/45 (84.4%), *p* = 0.01], but also in 60.5% (23/38) of PAK lesions. A pigmented network was present in the vast majority of PAK lesions [10/45 of NPAK (22.2%), 34/38 of PAK (89.5%), *p* < 0.001].

Concerning the pigment intensity, even less than 10% was significantly associated with PAK (*p* = 0.001). Scales were equally observed in NPAK and PAK lesions [38/45 (84.4%), 32/38 (84.2%), respectively]. The structures that were equally recorded in both groups of lesions were widened follicular openings, yellowish dots, and white circles (77.9% NPAK vs. 76.3% PAK, 62.2% NPAK vs. 63.2% PAK, 62.2% NPAK VS. 63.2% PAK, respectively).

Consequently, linear wavy vessels and shiny streaks were more frequently revealed in NPAK lesions (66.7% and 37.8% in NPAK, 57.9% and 29% in PAK, respectively). On the contrary, rosettes and a red starburst pattern were more commonly observed in PAK lesions (42.1% and 21% vs. 31.1% and 8.9%, respectively) (Figure 1).

### 3.3. Diagnostic Significance of Dermoscopic Structures

For statistically significant dermoscopic features with *p* < 0.05 of NPAK and PAK, Se, Sp, PPV, and NPV were calculated (Table 3). Among the dermoscopic results of NPAK, the red network had high Se (84%) but low Sp (39%) values, while the pigmented network and pigment intensity less than 10% had high Se (89% and 83%, respectively), Sp (77% for both), and NPV values (89% and 93%, respectively). When the pigment intensity was greater than 10%, Se, Sp, PPV, and NPV values were high (90%, 86%, 79%, and 93%, respectively).

Finally, the results of the logistic regression analysis are presented in Table 4. The odds ratio for the red pseudo-network is 0.28, 29.75 for pigmented pseudo-network, 17.22 for pigment intensity less than 10%, and 58.9 for pigment intensity greater than 10%.

## 4. Discussion

In this study, we examined the demographic, clinical, and dermoscopic characteristics aiming to detect special observations that could accurately differentiate NPAK from PAK.

Concerning age, gender, and skin type according to the Fitzpatrick classification of the participant, no significant differences were observed between NPAK and PAK lesions. Patients of both groups tended to be of an older age, with a preponderance of the male sex, and observations in agreement with previous epidemiological studies [5,17]. In the NPAK group, skin phototype II was predominant, while in the PAK group, phototype III was more common. Phototype I was rarely observed in PAK compared to NPAK, a result consistent with other reports, suggestive of the tendency of PAK to present darker phototypes [1].

Moreover, results regarding the macroscopic examination of lesions revealed no statistical significance. The majority (approximately one-half) of NPAK were located in the scalp, and less than one-quarter in the lateral face, while PAK was equally distributed in the lateral face, scalp, and temples. Our observation was in line with the previous data [17]. In addition, according to Olsen’s clinical grading, NPAK and PAK displayed similar frequencies in stages I to III, as mentioned in the existing literature [18].

The dermoscopic examination could confidently differentiate NPAK from PAK through color. A pigmented pseudo-network is a potent predictor of PAK (22.2% NPAK vs. 89.5% PAK) demonstrating high Se and NPV values, moderate Sp and PPV values, and an odds ratio of 29.75 for pigmented actinic lesions in accordance with the previous reports [19]. Pigment intensity of more than 10% had discriminating efficiency for PAK versus NPAK with high Se and NPV, moderate Sp and PPV, and an odds ratio of 58.9. Pigment intensity of less than 10% was also statistically significant with high NPP, moderate Se and Sp, low PPV, and an odds ratio of 17.22. The presence of a red pseudo-network was significantly predominant in NPAK lesions (84.4% NPAK vs. 60.5% PAK) but without discriminative potential (moderate Se, low PPV, and NPV, very-low Sp, odds ratio of 0.28), in concordance with the previous dermoscopic data [15]. The detection of a red pseudo-network in PAK could be correlated with the presence of significant histological inflammation, which is a characteristic of these pre-malignant lesions [20].

Finally, the presence of scale, targetoid follicular openings, rosettes, linear wavy vessels, and shiny streaks did not meet statistical importance for the differentiation between NPAK and PAK, since these features were observed at almost similar frequencies in both types of lesions. Interestingly, the red starburst pattern, a sign of more aggressive lesions, was close to statistical significance for PAK (8.9% NPAK vs. 21% PAK, *p* = 0.13).

The limitations of our study were the relatively small number of examined lesions, the non-blinded selection of patients and lesions, and the fact that the investigators were aware of the diagnosis. In addition, the accuracy of the dermoscopic criteria referred only to the differentiation of NPAK from PAK, and therefore could not be generalized. Therefore, further studies should be conducted, including non-pigmented and pigmented skin lesions with similar macroscopic presentations, in order to observe special AK dermoscopic features.

## 5. Conclusions

Our study determined that pigmentation is the main variable that could clinically and dermoscopically differentiate NPAK from PAK. Regarding other well-established AK dermoscopic features (i.e., scale, targetoid follicular openings, rosettes, linear wavy vessels, and shiny streaks), these are evenly observed among NPAK and PAK failing to show statistical significance. The presence of a pigmented pseudo-network with a pigment intensity of more than 10% is an accurate, statistically important dermoscopic clue for the differential diagnosis between NPAK and PAK.

## Figures and Tables

**Figure 1 jcm-12-01063-f001:**
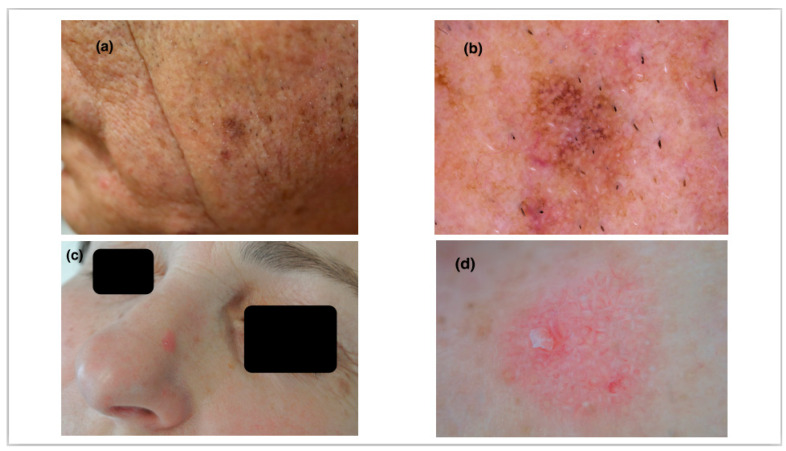
(**a**) A 67-year-old male patient with a pigmented scaly plaque on the left lateral side of the face. (**b**) Pigmented pseudo-network and widened follicular openings along with fine scaling demonstrate a typical pigmented actinic keratosis of the face. (**c**) A 62-year-old female patient with a pink, slightly elevated plaque on the nose. (**d**) A red pseudo-network surrounding widened follicular ostia and yellowish dots are features of a non-pigmented actinic keratosis. Linear wavy vessels can also be observed.

**Table 1 jcm-12-01063-t001:** Demographic and clinical characteristics of the study’s population.

	NPAK 45 (54.2%)	PAK 38 (45.8%)	*p*
Age (mean) ± SD, y.	78.7 ± 11.5	80.6 ± 7.9	0.41
Sex (male)	38 (84.4)	31 (81.6)	0.73
Fitzpatrick phototype			
I	8 (17.8)	2 (5.3)	0.46
II	17 (37.8)	15 (39.5)	
III	13 (28.9)	16 (42.1)	
IV	7(15.5)	5(13.1)	
Lesion location			
Scalp	18 (40)	10 (26.3)	0.19
Forehead	8 (17.8)	2 (5.3)	
Temples	7 (15.6)	10 (26.3)	
Lateral face	9 (20)	11 (28.9)	
Nose	3 (6.6)	4 (10.6)	
Ear		1 (2.6)	
Olsen’s clinical stage			
I	16 (35.6)	13 (34.2)	0.74
II	18 (40)	18 (47.4)	
III	11 (24.4)	7 (18.4)	

NPAK: non-pigmented actinic keratosis; PAK: pigmented actinic keratosis.

**Table 2 jcm-12-01063-t002:** Dermoscopic features of NPAK and PAK.

	NPAK 45 (54.2%)	PAK 38 (45.8%)	*p*
Color			
Red pseudo-network (%)	38 (84.4)	23 (60.5)	0.01
Pigmented pseudo-network (%)	10 (22.2)	34 (89.5)	<0.001
Pigment intensity <10%	9 (20)	10 (26.3)	0.001
Pigment intensity 10–50%	5 (11.1)	19 (50)	<0.001
Pigment intensity >50%	0 (0)	7 (18.4)	N/A
Scale	38 (84.4)	32 (84.2)	0.98
Follicular openings			
Widened follicular openings	35 (77.9)	29 (76.3)	0.87
Yellowish dots	28 (62.2)	24 (63.2)	0.93
White circles	28 (62.2)	24 (63.2)	0.93
Rosettes	14 (31.1)	16 (42.1)	0.3
Linear wavy vessels	30 (66.7)	22 (57.9)	0.41
Shiny streaks	17 (37.8%)	11 (29%)	0.4
Red starburst	4 (8.9%)	8 (21%)	0.13

NPAK: non-pigmented actinic keratosis; PAK: pigmented actinic keratosis.

**Table 3 jcm-12-01063-t003:** Se, Sp, PPV, and NPV of significantly present dermoscopic criteria for NPAK and PAK.

	Se	Sp	PPV	NPV
NPAK				
Red pseudo-network	84%	39%	62%	68%
PAK				
Pigmented pseudo-network	89%	77%	77%	89%
Pigment intensity <10%	83%	77%	52%	93%
Pigment intensity 10–50%	90%	86%	79%	93%

Se: sensitivity; Sp: specificity; PPV: positive predictive value; NPV: negative predictive value; NPAK: non-pigmented actinic keratosis; PAK: pigmented actinic keratosis.

**Table 4 jcm-12-01063-t004:** Logistic regression analysis of statistically significant dermoscopic criteria.

	OR	95% Cl	*p*
NPAK vs. PAK			
Red pseudo-network	0.28	0.1–0.8	0.02
Pigmentedpseudo-network	29.75	8.51–104.03	<0.001
Pigment intensity vs. none			
<10%	17.22	3.18–93.32	0.001
10–50%	58.9	10.37–334.37	<0.001
>50%			N/A

NPAK: non-pigmented actinic keratosis; PAK: pigmented actinic keratosis.

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
