# Peer review of "Dermoscopy of Actinic Keratosis: Is There a True Differentiation between Non-Pigmented and Pigmented Lesions?"

_jcm, 2023, doi:10.3390/jcm12031063_

Round 1
Reviewer 1 Report
The study is focused in finding hoe are pigmented and non-pigmented actinic keratosis (AK) distinguished based in dermoscopy.
The design of the study is simple and correct regarding the design, the material, and methods. The authors translate the exploration into numbers and then analyzed them obtaining a well-designed database. With respect to that, I would like to know how the dermoscopic data were compiled are they are subjective anyway and that the authors assessed the correlation between the two observer investigators (DS and AK).
The results are coherent, and the discussion is correct.
Author Response
We thank the reviewer for the valuable comments. We agree with the reviewer’s revisions.
- Please check underlined text in lines 78-80.
In specific: All the lesions evaluated by the two investigators (DS and AK) met agreement for their dermoscopic criteria. When the investigators’ assessment was not in accordance lesions were excluded from the study.
Reviewer 2 Report
A very interesting study, presented in a clear way.
1.Does it have any clinical significance to differentiate between pigmented and non pigmented AK? Does the management/treatment differ according to the AK subtype? Please comment on this in the introduction or discussion
2. How did you measure the pigment intensity? Was it the % of lesion surface?
3. As you are using the Olsen's scale in Results, please include it and briefly characterize in the Methods section
Author Response
We thank the reviewer for the valuable comments. We agree with the reviewer’s revisions and suggestions.
In specific:
- The differentiation between pigmented and non-pigmented actinic keratosis does not actually seem to have major clinical significance at first sight. However, our study aims to investigate aspects of the dermoscopic morphology of AKs seen from the alternative perspective of the comparison between pigmented and non-pigmented clinical subtypes. Pigmented AKs represent a diagnostic pitfall in everyday clinical practice due to their clinico-dermoscopic similarity to several benign or malignant lesions, such as seborrheic keratoses or lentigo maligna, respectively. Despite their similarity these clinical entities have a completely different therapeutic approach. Given also that PAKs are very commonly observed on sun-exposed sites in darker skin phototypes it is a major challenge to describe all possible dermoscopic characteristics of them.
Please check the underlined text in lines 67-71 in the “introduction” section.
- The intensity of the pigmentation was measured as the percentage of the lesion’s surface covered by the pigment.
Please check the underlined text in lines 99-100
- As correctly suggested by the reviewer we describe Olsen’s scale in the “methods” section and specifically in lines 90-93 (underlined text).